# Three-Dimensional Landing Zone Segmentation in Urbanized Aerial Images from Depth Information Using a Deep Neural Network–Superpixel Approach

**DOI:** 10.3390/s25082517

**Published:** 2025-04-17

**Authors:** N. A. Morales-Navarro, J. A. de Jesús Osuna-Coutiño, Madaín Pérez-Patricio, J. L. Camas-Anzueto, J. Renán Velázquez-González, Abiel Aguilar-González, Ernesto Alonso Ocaña-Valenzuela, Juan-Belisario Ibarra-de-la-Garza

**Affiliations:** 1Department of Science, Tecnológico Nacional de México/IT de Tuxtla Gutiérrez, Carr. Panamericana Km. 1080, Tuxtla Gutiérrez 29050, Chiapas, Mexico; nestor.mn@tuxtla.tecnm.mx (N.A.M.-N.); antonio.osuna@tuxtla.tecnm.mx (J.A.d.J.O.-C.); jcamas@ittg.edu.mx (J.L.C.-A.); josue.velazquez@ittuxtlagutierrez.edu.mx (J.R.V.-G.); 2Department of Computer Science, Instituto Nacional de Astrofísica, Óptica y Electrónica, Luis Enrique Erro No. 1, Santa María Tonantzintla 72840, Puebla, Mexico; abiel@inaoep.mx; 3Department of Mechatronics, Tecnológico Nacional de México/ITS de Comalcalco, Carretera Vecinal, Comalcalco-Paraíso Km. 2, Comalcalco 86651, Tabasco, Mexico; ernesto.ocana@comalcalco.tecnm.mx; 4Department of Systems and Computing, Tecnológico Nacional de México/IT de Linares, Carr. Nacional Km. 157, Linares 67700, Nuevo León, Mexico; jibarra@linares.tecnm.mx

**Keywords:** three-dimensional semantic segmentation, landing zone detection, superpixel segmentation, deep learning

## Abstract

Landing zone detection of autonomous aerial vehicles is crucial for locating suitable landing areas. Currently, landing zone localization predominantly relies on methods that use RGB cameras. These sensors offer the advantage of integration into the majority of autonomous vehicles. However, they lack depth perception, which can lead to the suggestion of non-viable landing zones, as they only assess an area using RGB information. They do not consider if the surface is irregular or accessible for a user (easily accessible to a person on foot). An alternative approach is to utilize 3D information extracted from depth images, but this introduces the challenge of correctly interpreting depth ambiguity. Motivated by the latter, we propose a methodology for 3D landing zone segmentation using a DNN-Superpixel approach. This methodology consists of three steps: First, the proposal involves clustering depth information using superpixels to segment, locate, and delimit zones within the scene. Second, we propose feature extraction from adjacent objects through a bounding box of the analyzed area. Finally, this methodology uses a Deep Neural Network (DNN) to segment a 3D area as landable or non-landable, considering its accessibility. The experimental results are feasible and promising. For example, the landing zone detection achieved an average recall of 0.953, meaning that this approach identified 95.3% of the pixels according to the ground truth. In addition, we have an average precision of 0.949, meaning that this approach segments 94.9% of the landing zones correctly.

## 1. Introduction

In computer vision, landing zone identification allows the Unmanned Aerial Vehicle (UAV) to land correctly and safely [1,2]. This task is crucial in urban areas, where it is essential to find flat and accessible landing areas to ensure the UAV’s stability during landing. Proper detection of these areas reduces the risk of structural damage or overturning of the UAV during descent, thus ensuring vehicle integrity [3]. Due to this benefit, several tasks use landing zone detection, including delivery services [4], urban search and rescue operations [5], and emergency landings [6].

There are several approaches to landing zone detection. One approach analyzes 3D point clouds using depth sensors. This analysis allows the detection of safe landing zones [7]. These methodologies typically adjust the geometric structures in a 3D point cloud. In addition, the geometric shape of each object is analyzed using statistical methods or deep learning algorithms for landing zone identification. However, current methods are based on heuristic rules [8] and simple geometric features to determine a landing zone [3]. These geometric approaches are susceptible to errors, particularly in green areas and environments with vegetation.

Another approach to landing zone detection involves identifying visual markers through deep learning algorithms [9,10]. This method analyzes images to recognize landing zones by detecting visual markers with predefined patterns. In addition, it does not provide information about the landing zone environment [1]. The effectiveness of this approach depends on placing the markers in visible locations, which can increase the risk of occlusion due to the UAV’s viewing perspective. Consequently, the placement and visibility of these markers can significantly impact the reliability of the landing zone detection.

Other researchers have implemented deep learning algorithms to detect safe landing zones in unknown environments [9,11,12]. These algorithms use Convolutional Neural Networks (CNNs) that can learn to detect landing zones based on their visual characteristics. Unlike other approaches, these algorithms have proven effective in image processing and have led to significant advances in object classification and detection. In most cases, a trained CNN can correctly generalize new images and scenarios. However, there are several challenges because these networks only consider RGB information. For example, it may suggest irregular or inaccessible landing zones by not taking depth information.

Motivated by the results of the learning algorithms and the benefits of interpreting depth images, this study aims to identify landing zones based on deep learning and depth information. In particular, we are interested in clustering depth information to infer 3D landing zones. This involves evaluating the three-dimensional environment by segmenting landing regions. Most studies focus exclusively on RGB information or visual markers without considering the context of the landing zone. As a result, by relying solely on color information, these approaches may fail in areas of homogeneous tonality. In addition, they do not include an assessment of terrain flatness nor analyze the presence of nearby objects that may pose a risk to the UAV during landing. For this reason, this study proposes a CNN–superpixel configuration that combines the abstraction power of deep learning and the clustering capacity using superpixels. This approach also takes into account the accessibility of the landing zones, making it useful for tasks like deliveries, rescue operations, and emergency landings. We consider that a landing zone is accessible when it is practical for a user to retrieve the drone easily, that is, free of obstacles and easily accessible to a person on foot.

In Section 2, we present related studies that determine the research location. The proposed method is described in Section 3. Section 4 describes the experiments designed to evaluate the feasibility of the proposed method and the obtained results. Finally, conclusions and future work are presented in the Section 5.

## 2. Related Works

This section provides a state-of-the-art overview of landing zone detection. In most cases, feature extraction from a UAV environment uses RGB and 3D sensors to determine the landing zone. For this purpose, we present two subsections on landing zone detection. The first section introduces the detection methodologies using RGB images (Section 2.1). The second section presents detection methodologies using 3D information (Section 2.2).

### 2.1. Landing Zone Detection with RGB Images

An alternative allows the extraction of visual marker features using threshold-based operations [13,14,15,16,17]. These techniques employ visual markers, typically a geometric shape, to identify landing zones. The threshold approach involves selecting a threshold value to create a binary image. In addition, morphological transformations are applied to the image to detect the centroid of the visual marking. The centroid contains the landing position of the UAV. This approach allows rapid detection of landing zones with a low computational cost. However, in outdoor environments, the marker can be confused with the environment, especially if there are similar surfaces, resulting in missed or false detection.

Various methods for detecting landing zones use learning algorithms such as Support Vector Machines (SVMs) or Artificial Neural Networks (ANNs) [18,19,20]. These studies detect zones or objects using RGB information and object shape, thus requiring prior knowledge of the features that describe each landing zone. These proposals combine artificial vision techniques with visual feature extraction, which allows the detection of a landing zone. These algorithms have the advantage of efficiency. However, these proposals do not train with as much data as deep learning (they do not generalize different environments). These algorithms work efficiently in specific scenarios but are susceptible to sudden lighting changes and scene rotation, making them unsuitable for use in uncontrolled environments.

Another alternative is to obtain the features of the landing zone using deep learning algorithms [10,21]. The motivation for these studies is that landing zones are usually detected using features with a more complex abstraction level. These algorithms implement Convolutional Neural Networks (CNNs) that can efficiently detect landing zones. Therefore, the algorithm requires a set of images that display landing zones for training. However, these methodologies lack depth information, which is a limitation because they can propose non-viable landing zones owing to the irregularity of their surfaces. In addition, these algorithms require a large training data set, resulting in longer training times. However, in most cases, the repositories do not include aerial images. This data limitation restricts the correct training of deep learning algorithms.

### 2.2. Landing Zone Detection with 3D Information

An alternative approach for identifying landing zones for UAVs involves statistical methodologies to perform plane fitting on three-dimensional point clouds [6,22,23,24]. These approaches have proven effective in assessing the terrain by UAVs, allowing the detection of flat areas. These methodologies use region-growing algorithms and Principal Component Analysis (PCA) to analyze these zones. Each region is analyzed using PCA by a plane adjustment and determines if a surface satisfies the requirements for the landing. However, they cannot differentiate critical contexts, such as landing on the roof of a building, in a garden, or on a street. This limitation underscores the need to integrate more complex and contextually relevant features to improve accuracy in landing zone identification in urban and natural environments.

Some authors combined three-dimensional data with two-dimensional information to address the challenge of segmenting and detecting landing zones in point clouds [25,26,27,28]. These methods extract visual features from 2D images, such as texture, color, and shape. These features provide information to identify areas or structures, such as buildings. Then, the 3D point cloud is calculated, and its geometric characteristics are analyzed to determine the viability of the terrain as a landing zone. However, these methodologies have the limitation of requiring a large volume of 2D and 3D images in different environments. In addition, the computational cost is considerably high, which presents an additional challenge in implementing these methodologies. The efficiency of these techniques is conditioned not only by the quality and quantity of the training data but also by the capability of the systems to handle intensive processing.

Another approach considers methodologies that identify the landing zone from a Convolutional Neural Network using a LiDAR sensor [29,30]. These methodologies propose a 3D Convolutional Neural Network (3DCNN) to determine the landing area. Subsequently, the CNN evaluates the areas by bounding boxes representing the landing zone with an extension of 1 to 3 m^3^. The 3DCNN technique is typically efficient in detecting landing zones in a point cloud. However, the 3DCNN requires a large amount of data for training, and some authors have opted to generate synthetic datasets and validate their methodologies using simulators. Furthermore, this approach usually has the limitation of not detecting large objects (buildings and trees) because relatively small areas are analyzed.

The proposed approach aims to address these problems in a novel way. This methodology relies on a single LiDAR or depth sensor without requiring additional sensors or external devices. The methodology identifies landing zones by interpreting depth images, allowing them to differentiate between a street and a building, even if they present similar geometrical characteristics. The methodology assesses each potential landing zone by considering the ease of recovering the UAV in each zone and detecting flat and accessible areas. An additional advantage is that our learning network requires a few images for training, as it can generate 875 samples per depth image. In addition, this study proposes a DNN-Superpixel configuration that combines the abstraction power of deep learning and the clustering capacity using superpixels.

## 3. Methodology

This section presents the proposed methodology for landing zone detection. Our strategy combines the abstraction power of deep learning with the information provided by the clustering of superpixels. In addition, this work considers the accessibility of landing zones with flat areas. This methodology consists of three phases: zone identification, feature extraction, and landing zone segmentation. First, the proposal involves clustering depth information using superpixels to segment, locate, and delimit zones within the scene (Section 3.1). Second, we propose feature extraction from adjacent objects through a bounding box of the analyzed area (Section 3.2.4). Finally, this methodology uses a Deep Neural Network (DNN) to segment an area as landable or non-landable, considering its accessibility (Section 3.3). A landing zone is accessible when it is practical for a user to retrieve the drone easily, that is, free of obstacles and easily accessible to a person on foot. Figure 1 shows the block diagram of the proposed methodology.

### 3.1. Zone Identification

In this section, zone identification involves dividing, locating, and delimiting zones in the scene. This process only uses the height information of the scene, unlike other methods. The proposed approach uses superpixel segmentation with depth information, which helps to eliminate issues such as object occlusion and changes in lighting that often cause problems in color images.

#### 3.1.1. Superpixel Segmentation

In image processing, region detection using superpixel segmentation divides an image into smaller segments known as superpixels. These segments are created by grouping nearby pixels with similar characteristics, such as color, texture, or intensity, to simplify and enhance image processing. When using RGB information, it may not provide sufficient characteristics to determine whether an area is flat or irregular, the height of adjacent areas, and whether it is accessible to landing. On the other hand, the complexity of using only depth images for zone detection is that features such as colors and textures that allow objects to be identified and differentiated are lost. However, segmenting superpixels in a depth image enables the detection, location, and delimitation of regions based on the height differences between objects found in the depth image.

We denote the depth image as $I_{d}$, which contains information on the heights of zones and objects observed by the UAV. The methodology uses superpixel segmentation $I_{d}$ and obtains the image with superpixel segmentation $I_{s}$. For this purpose, we evaluate the behavior of the superpixel segmentation algorithms Felzenszwalb [31], QuickShift [32], and SLIC [33] using depth images. Our evaluation demonstrates that QuickShift produces the best results. Thus, this methodology uses the QuickShift superpixel approach, where $\phi_{i}$ denotes the *i*-th superpixel in image $I_{s}$. Each segment contains a set of pixels with similar heights.

Therefore, each $\phi_{i}$ has a set of heights that are distinct from the other adjacent segments. This methodology uses the superpixels as the possible landing zones. For the QuickShift approach, we use the following parameters: window size for density estimation = 3, maximum allowed distance between adjacent point = 6, and shape smoothing term = 0.01. These parameters are standard for all processed images and provide segments with an area sufficiently large for landing tasks. Figure 2 shows ten superpixels in $I_{s}$, each representing areas or objects (trees, buildings, ground, among others).

A depth image lacks information such as color and texture, making it difficult to identify objects or areas. In our experiments, the QuickShift algorithm demonstrated better effectiveness in segmenting superpixels using depth images, achieving an efficiency of 94.9%. The above is because QuickShift tends to provide better results, due to its ability to adapt more flexibly to variations in intensity and texture, compared to SLIC, which achieved an efficiency of 76.74%, and Felzenszwalb, with a performance of 62.32%. QuickShift is particularly effective in preserving edges and complex structures using depth images, where fine details are essential. On the other hand, SLIC, although efficient, can lose detail in regions with smooth gradients. Felzenszwalb, although robust, tends to over-segment in scenarios with heterogeneous textures, which makes it less suitable for this type of image.

### 3.2. Feature Extraction of Potential Landing Zones

This subsection focuses on extracting features used as inputs to the deep learning network. The methodology first analyzes whether the superpixel corresponds to a plane or an irregular zone. The height of each superpixel is then homogenized to simplify the analysis by obtaining a mean value. Finally, the process extracts features of adjacent superpixels to identify abrupt changes in depth between them.

#### 3.2.1. Pinhole Model

The pinhole camera model focuses on the projection of light reflected from a point in space p(x,y,z) towards a point in the image plane p(x,y) that passes through the camera center $C_{o}$. The optical center, denoted as $O_{o}$, is the origin of the coordinates in the image plane. In practice, the actual position of the optical center may not align with the geometric center of the image plane [34]. This depends on the intrinsic parameters of the camera. The focal length *f* is the distance between the optical center $O_{o}$ and the image plane. Therefore, we can calculate which point p(x,y,z) maps to point $p(\frac{fX}{z},\frac{fY}{z},f)$ on the image plane employing similar triangles, as seen in Figure 3.

The methodology uses the pinhole model with the image plane information and the estimated depth to obtain the 3D point cloud representing the surface of each superpixel. We divide the scale factor *k* with a value of 255 (maximum intensity in an eight-bit image) and multiplied it by the depth information *z* to convert the data in meters. Scale factor *k* is the maximum depth we can obtain. For example, considering a maximum depth of 300 m and a grayscale of 80 on the depth image, *Z* is approximately 94.11 m. Equations (Equation 1)–(Equation 3) calculate the coordinates (X,Y,Z) of a point in the space.(1)$$Z=\frac{k\times z}{255}$$(2)$$X=\frac{x\times Z}{f}$$(3)$$Y=\frac{y\times Z}{f}$$

#### 3.2.2. Principal Component Analysis

This methodology obtains the error between a fit plane and point cloud using Principal Component Analysis (PCA). This error allows us to evaluate whether the surface is flat or irregular. Algebraically, PCA is a particular linear combination of an original variable set, and geometrically, this is equivalent to the rotation of the original coordinate system to a new one, where the new axes represent the maximum variability of the dataset [35]. For this purpose, we denote the fitting plane of the point cloud as *V*, which is a point cloud of *N* points denoted as a 3×N matrix that corresponds to a possible landing zone represented by superpixel $\phi_{i}$, where $\left(X_{i},Y_{i},Z_{i}\right)$ are the coordinates of a particular point within point cloud *V*. Therefore, we first obtain the mean values $\bar{V}$ of point cloud *V* using Equations (Equation 4)–(Equation 6) to find the fitting plane.(4)$$\bar{X}=\frac{1}{N}\sum_{i=1}^{N}X_{i}$$(5)$$\bar{Y}=\frac{1}{N}\sum_{i=1}^{N}Y_{i}$$(6)$$\bar{Z}=\frac{1}{N}\sum_{i=1}^{N}Z_{i}$$

Subsequently, we obtained the covariance matrix *C* from the linear combination of the dimensions in (X,Y,Z) of *V* (see Equation (Equation 7)), where $C_{q,q}\prime =\frac{1}{N-1}\sum_{i=1}^{N}\left(q_{i}-\bar{q}\right)\left(q_{i}\prime -\bar{q\prime }\right)$, with q={x,y,z} and $\bar{q}=\left\{\bar{x},\bar{y},\bar{z}\right\}$ as the mean.(7)$$C=\left[\begin{array}{ccc} C_{x,x} & C_{y,x} & C_{z,x}\\ C_{x,y} & C_{y,y} & C_{z,y}\\ C_{x,z} & C_{y,z} & C_{z,z} \end{array}\right]$$

After obtaining the covariance matrix *C*, the methodology calculates the eigenvalues λ using the Equation (Equation 8), where λ is the characteristic root of the determinant equation and *I* is an identity matrix of order 3×3 [36].(8)|C−λI|=0

Equation (Equation 8) is a polynomial of degree 3 with respect to λ, and therefore it has three roots that can be ordered, as shown in Equation (Equation 9).(9)$$\lambda_{n}\geq \lambda_{a}\geq \lambda_{b}$$

Subsequently, for each eigenvalue $\lambda_{n}$, $\lambda_{a}$, $\lambda_{b}$ the methodology obtains its corresponding eigenvector $e_{n}$, $e_{a}$, $e_{b}$. Therefore, we obtained each vector $e_{i}$ using Equation (Equation 10) [36].(10)$$\left(C-\lambda I\right)e_{i}=0$$

Afterward, the methodology obtains a matrix *E* composed of the eigenvectors $e_{n}$, $e_{a}$, $e_{b}$, as expressed in Equation (Equation 11).(11)$$C=\left[\begin{array}{ccc} E_{n,x} & E_{n,x} & E_{n,x}\\ E_{a,y} & E_{a,y} & E_{a,y}\\ E_{b,z} & E_{b,z} & E_{b,z} \end{array}\right]$$

Then, we obtain the matrix $V^{\prime }=\left(X^{\prime },Y^{\prime },Z^{\prime }\right)$ using Equation (Equation 12), which contains the new set of *N* points corresponding to the fitting plane [35].(12)$$V^{\prime }=E^{-1}V-\bar{V}$$

Finally, the methodology calculates the mean square error (*MSE*) between the 3D coordinates of the sets of points *V* and $V^{\prime }$ using Equation (Equation 13).(13)$$MSE=\frac{1}{n}\sum_{i=1}^{n}{\left(V_{i}-V_{i}^{\prime }\right)}^{2}$$

The *MSE* provides a value of how much the plane fits $V^{\prime }$ into the point cloud *V*. Therefore, a value close to zero indicates that the analyzed zone (superpixel) can be considered flat. Otherwise, the zone must classified as irregular. The methodology used a threshold value of 0.4 to evaluate the *MSE* and determine whether the zone is classified as flat or irregular. We perform this evaluation using Equation (Equation 14).(14)$$f\left(MSE\right)=\left\{\begin{array}{cc} ‘‘Flat’’ & \mathrm{si}\;MSE\leq 0.4\\ ‘‘Irregular’’ & \mathrm{si}\;MSE>0.4 \end{array}\right.$$

In the proposed methodology, a landing zone must be sufficiently large, flat, and accessible to the user to pick up the UAV. Therefore, these areas (superpixel ϕ) considered flat are analyzed using a DNN to evaluate their accessibility for a user to pick up the UAV and classify it as landable or non-landable.

#### 3.2.3. Height Homogenization

We obtain a set of heights for each superpixel $\phi_{i}$ of image $I_{s}$, which we denote as $\phi_{i}=\left\{\psi_{0},\psi_{1},\cdots ,\psi_{n}\right\}$$\in I_{s},$ where *n* represents the *n*-th pixel contained within the boundary of a superpixel $\phi_{i}$, ψ is the height contained in each pixel, ψ(x,y) is the coordinate, *x* is the abscissa, and *y* is the ordinate. In addition, this methodology obtains one homogenized superpixel $\phi_{i}^{\prime }$ for each analyzed superpixel $\phi_{i}$. Then, the process calculates the mean value of the sets ϕ to homogenize the superpixel height (Equation (Equation 15)). Finally, this process groups the superpixels $\phi_{i}^{\prime }$ to obtain images $I_{S}^{\prime }$.(15)$$\phi_{i}^{\prime }=\frac{1}{n}\sum_{j=1}^{n}\psi_{j}$$

#### 3.2.4. Feature Extraction

A superpixel lacks a standard size and shape, resulting in a variable number of neighboring superpixels. We extract nine features that allow us to analyze each zone (superpixel). These features are composed of the height of the homogenized superpixel $\phi_{i}^{\prime }$ and the heights of the adjoining homogenized superpixels. The proposal uses a bounding box to localize and standardize the number of features. The bounding box is a rectangular region surrounding a superpixel, and its boundaries are formed by the upper left corner $\left(x_{0},y_{0}\right)$ and the lower right corner $\left(x_{1},y_{1}\right)$. The methodology obtains the features from nine coordinates extracted from the bounding box (see Figure 4): $\left(x_{0},y_{0}\right),\left(x_{2},y_{0}\right),\left(x_{1},y_{0}\right),\left(x_{0},y_{2}\right),\left(x_{2},y_{2}\right),\left(x_{1},y_{2}\right),\left(x_{0},y_{1}\right),$ $\left(x_{2},y_{1}\right),\left(x_{1},y_{1}\right)$. We use Equations (Equation 16) and (Equation 17) to obtain the values of $x_{2}$ and $y_{2}$.(16)$$x_{2}=\frac{x_{1}-x_{0}}{2}$$(17)$$y_{2}=\frac{y_{1}-y_{0}}{2}$$

The methodology obtains homogenized heights from the coordinates extracted from the bounding box. Nine features are extracted in two steps. The first step involves extracting the initial feature from the homogenized height of the superpixel under analysis, referred to as $\rho =I_{S}^{\prime }\left(x_{2},y_{2}\right)$. Subsequently, the remaining eight features are obtained from the difference between the homogenized heights at the limit of the bounding box and ρ. We denote the set of features as vector $H_{i}$ (Equation (Equation 18)). Figure 4 shows an example of nine homogenized superpixels.(18)$$H_{i}=\begin{array}{ccc} {}[I_{s}^{\prime }\left(x_{0},y_{0}\right)-\rho , & I_{s}^{\prime }\left(x_{2},y_{0}\right)-\rho , & I_{s}^{\prime }\left(x_{1},y_{0}\right)-\rho ,\\ I_{s}^{\prime }\left(x_{0},y_{2}\right)-\rho , & I_{s}^{\prime }\left(x_{2},y_{2}\right), & I_{s}^{\prime }\left(x_{1},y_{2}\right)-\rho ,\\ I_{s}^{\prime }\left(x_{0},y_{1}\right)-\rho , & I_{s}^{\prime }\left(x_{2},y_{1}\right)-\rho , & I_{s}^{\prime }\left(x_{1},y_{1}\right)-\rho ] \end{array}$$

In this methodology, the superpixel analyzed (at position $I_{S}^{\prime }\left(x_{2},y_{2}\right)$) must have adjacent superpixels around it. Therefore, this methodology cannot analyze the superpixels at the edges of the image. This is because the superpixels do not have adjacencies in some quadrants, and by not having these characteristics, our network could generate errors in the classification of the areas (Section 3.3). The methodology allows the detection of landing zones from a few depth images. For example, we can obtain up to 875 vectors *H* in each depth image $I_{d}$. Therefore, this methodology allows us to obtain a training dataset of 43,750 landing zones from a small set of 50 depth images.

### 3.3. Landing Zone Segmentation

We use a Deep Neural Network (DNN) to train and segment landing zones. Initially, the methodology trains the network by analyzing the features of zones and classifying them as landable or non-landable. For training, we utilize a superpixel and its adjacent superpixels (Section 3.2.4). These features help us identify sudden depth changes between these areas and assess whether the zone is accessible. This accessibility ensures easy UAV retrieval by the user.

#### 3.3.1. Training Dataset

In the training set for area classification, the methodology used aerial images captured by a UAV at altitudes of approximately 300 m (VPAIR [37]), 50 m (VALID [38]), and 26 m (ITTG datasets). The training datasets consist of aerial images with depth information from different outdoor environments. To construct the training set, we partition the depth image $I_{d}$ into superpixels $I_{s}$ (Section 3.1.1), with each superpixel representing a possible landing zone. Subsequently, we homogenize the height of each superpixel to simplify the analyzed information by obtaining a mean value (Section 3.2.3). Finally, we extract features of adjacent superpixels to identify abrupt changes in depth between them (Section 3.2.4).

With this feature extraction method, this approach can generate a large dataset from a small number of images. For example, we can obtain approximately 83,125 training landing zones on 95 images, meaning this approach can find 875 training landing zones in a single image (1024 × 1024 pixels). This dataset enables us to train a DNN and leverage the abstraction capabilities of deep learning to achieve robust performance for complex tasks such as segmenting landing zones. In addition, the proposed methodology considers accessibility zones and excludes hard-to-reach areas from the dataset. For example, we discard building roofs or tall flat structures during training.

#### 3.3.2. Accessible Landing Zone

This methodology analyzes whether the landing zone is accessible. We consider that a landing zone is accessible when it is practical for a user to retrieve the drone easily, that is, free of obstacles and easily accessible to a person on foot. For this purpose, this methodology evaluates if the landing area (superpixel) corresponds to a plane or irregular zone (Section 3.2.2). In addition, we consider accessibility zones while excluding hard-to-reach areas from the dataset (Section 3.3.1); that is, we discard building roofs or tall flat structures during training.

#### 3.3.3. DNN for Landing Zone Segmentation

We performed an exhaustive exploration of DNN architectures to minimize the classification errors in the proposed methodology. We carried out different combinations of hidden layers and neurons. Through this process, an efficient and optimal DNN architecture emerged, characterized by one input layer (9), five hidden layers (128, 64, 32, 64, 128), and one output layer (2). Figure 5 shows our architecture for zone classification.

Our DNN has an input layer consisting of nine neurons. These neurons use the characteristic vector $H_{i}$ which corresponds to nine areas. Then, the hidden layers adjust the weights between the neural connections to perform predictions to minimize the difference to the correct labels. This allows the DNN to learn and adapt to the complexity of the input data. The output layer consists of two neurons that classify the evaluated areas into two classes using a sigmoid activation function.

We use the sigmoid function in the output layer because it is suitable for binary classification problems; that is, this methodology classifies only two classes. The sigmoid function is represented by Equation (Equation 19), where $y_{i}$ and $y_{i-1}$ represent the outputs of the two connected layers. The sigmoid function transforms the values entered into a (0,1) scale, where 1 indicates that the zone is landable and 0 indicates that the zone is non-landable, as seen in Equation (Equation 20). We consider different colors to display the labels (blue for landable and red for non-landable).(19)$$f\left(y_{i}\right)=sigmoidal\left(\frac{1}{1-e^{-y_{i-1}}}\right)$$(20)$$sigmoidal\left(x\right)=\left\{\begin{array}{c} 1,x>0\\ 0,x\leq 0 \end{array}\right.$$

The loss function provides a prediction error between the actual and desired outputs. The weights and biases are adjusted to minimize the loss function during DNN training. This process uses optimization algorithms such as gradient descent, which searches for the direction in which the loss function decreases most rapidly. Gradient descent is one of the most popular algorithms for optimizing loss functions in neural networks. We compared various gradient algorithms to minimize the loss function. We used the adaptive moment estimation (Adam) since it demonstrated superior results due to its rapid convergence.

### 3.4. Diagram of Our Methodology

The proposed methodology, illustrated in Figure 6, is divided into three main phases. The first phase focuses on zone identification by segmenting the depth image using the QuickShift superpixel method. In the second phase, features are extracted from the zones, which include generating the 3D point cloud of each zone and assessing the flatness of the surface using principal component analysis (PCA). If the surface is flat, we proceed with height homogenization and additional feature extraction; non-flat surfaces are considered non-landable. Finally, in the third phase, zone segmentation is performed using a Deep Neural Network (DNN) that classifies each zone as landable or non-landable.

## 4. Discussion

In this section, we present the experiments of landing zone detection. We realize the quantitative evaluations using recall, precision, specificity, Jaccard index, accuracy, F1-score, Area Under the Curve, and PAM measures (Section 4.1). We compare several approaches (Principal Component Analysis, Support Vector Machines, Logistic Regression, YOLOv7, and our DNN-Superpixel Net) for landing zone segmentation (Section 4.2). Finally, the results of the approaches are discussed using evaluation metrics (Section 4.3 and Section 4.4).

### 4.1. Metrics

The proposed methodology uses the VPAIR [37], VALID [38], and ITTG (proposed) datasets to evaluate the classification of the landing zones. For this purpose, we manually labeled the ground truth of the dataset. This ground truth consists of two possible labels (landing and non-landing zones). We consider different colors to display the labels (blue for landing zones and red for non-landing zones).

Quantitative evaluation was performed using eight measures: recall (RE), precision (PR), specificity (SP), jaccard index (JI), accuracy (AC), F1-score (F1), Area Under the Curve (AUC), and Polygon Area Metric (PAM) based on the number of true positives, true negatives, false positives, and false negatives. The true positives, TPs, count the pixels of images whose segmentation was predicted correctly concerning the ground truth. To count the number of true negatives, TNs, we proceed as follows: Suppose that we are interested in the landing label. Then, all those pixels corresponding to other label rather than landing, according to the ground truth, should have received any other predicted label except the landing label; if that is the case, each of these pixels is counted as true negatives. False positives, FPs, correspond to all pixels with incorrect labels. Finally, false negatives, FNs, correspond to those pixels that should have received a specific label, but the prediction did not assign it correspondingly; for instance, those pixels corresponding to landing should have received a landing label. However, if any image did not receive such a label, those are counted as false negatives, FNs.

We assess precision using a standard metric commonly employed in image segmentation tasks. This metric is the ratio of true positives (TPs) to the sum of true positives (TPs) and false positives (FPs) (see Equation (Equation 22)). Unlike precision measured in terms of absolute error or probabilistic uncertainty, in this context, precision reflects the model’s ability to minimize false positives when identifying landing zones. Unlike precision calculated in terms of absolute error or probabilistic uncertainty, in this case, precision is interpreted as the ability of the model to minimize false positives in the identification of landing zones.

We used recall (RE) to measure the proportion of pixels whose respective labels were predicted correctly regarding the number of pixels in the ground truth. In simple terms, it is the ground truth that was correctly predicted. Precision (PR) is the proportion of pixels that were segmented correctly, that is, considering our segmentation. We employed specificity (SP) to measure the proportion of true negatives correctly identified in all negative cases, that is, the probability that the test is classified as negative when it is negative. The Jaccard index (JI) is utilized to evaluate the similarity and diversity of the predicted labels compared to the ground truth labels. The Jaccard index is calculated as the number of values belonging to both sets (intersection) divided by the unique number across both sets (union).

Accuracy (AC) is the proportion of correct predictions (TPs and TNs) divided by the number of examined cases. The F1-score (F1) helps summarize the performance of the predictions returned by the system. For a system with good performance, both recall and precision should tend to be one, meaning that most of the system’s predictions tend to be correct and that such predictions tend to cover most of the ground truth.(21)$$\mathrm{recall}\left(\mathrm{RE}\right)/\mathrm{sensitivity}\left(\mathrm{SE}\right)=\frac{\mathrm{Tp}}{\mathrm{Tp}+\mathrm{Fn}}$$(22)$$\mathrm{precision}\left(\mathrm{PR}\right)=\frac{\mathrm{Tp}}{\mathrm{Tp}+\mathrm{Fp}}$$(23)$$\mathrm{specificity}\left(\mathrm{SP}\right)=\frac{\mathrm{Tn}}{\mathrm{Tn}+\mathrm{Fp}}$$(24)$$\mathrm{Jaccard}\;\mathrm{index}\left(\mathrm{JI}\right)=\frac{\mathrm{Tp}}{\mathrm{Tp}+\mathrm{Fp}+\mathrm{Fn}}$$(25)$$\mathrm{accuracy}\left(\mathrm{AC}\right)=\frac{\mathrm{Tp}+\mathrm{Tn}}{\mathrm{Tp}+\mathrm{Tn}+\mathrm{Fp}+\mathrm{Fn}}$$(26)$$F1=2\frac{\mathrm{recall}\times \mathrm{precision}}{\mathrm{recall}+\mathrm{precision}}=\frac{\mathrm{Tp}}{\mathrm{Tp}+\frac{1}{2}\left(\mathrm{Fp}+\mathrm{Fn}\right)}$$

The Area Under the Curve (AUC) provides a measure of the model to discriminate between the classes. This metric represents the area under the receiver operating characteristic (ROC) curve, which plots the true positive rate against the false positive rate at various threshold settings, where f(x) is a receiver operating characteristic curve in which the true positive rate (SE) is plotted in the function of the false positive rate (1-SP) for different cut-off points. The Polygon Area Metric (PAM) [39] is calculated by determining the area of the polygon formed by the points representing RE/SE, SP, JI, AC, F1, and AUC within a regular hexagon. It is important to note that the regular hexagon consists of six sides, each with a length of 1, and the total area of the hexagon is 2.59807. The lengths from the center towards the hexagon vertex correspond to the values of RE/SE, SP, JI, AC, F1, and AUC, respectively, where PA represents the area of the formed polygon. It is important to mention that to normalize the PAM within the [0, 1] range, the PA value is divided by 2.59807.(27)$$\mathrm{AUC}=\int_{0}^{1}f\left(x\right)\;\mathrm{dx}$$(28)$$\mathrm{PAM}=\frac{\mathrm{PA}}{2.59807}$$

### 4.2. Landing Segmentation Approaches

We compared six approaches (Principal Component Analysis, Support Vector Machines, Logistic Regression, YOLOv7-RGB, YOLOv7-Depth, and DNN-Superpixel Net) for landing zone detection. Principal Component Analysis (PCA) involves analyzing landing zones to determine their flatness through statistical analysis [22,40]. Support Vector Machines (SVMs) is a supervised learning algorithm that handles nonlinear data and uses kernels to map features to a larger space to linearize the data [41]. Logistic regression is a technique that assumes a linear relationship between features and landing zones [42]. Yolov7-RGB is a CNN that extracts hierarchical features from an RGB image by predicting through segmentation masks a set of pixels that belong to a class within a bounding box [43]. In addition, we use YOLOv7 to predict landing zones using only depth images, which we call YOLOv7-Depth. Finally, DNN-Superpixel Net is our proposed for landing zone segmentation using depth information.

### 4.3. Segmentation Result

PCA involves analyzing landing zones to determine their flatness through statistical analysis. However, unlike other approaches, this technique does not analyze the area environment to be evaluated. This approach could provide incorrect classification, reducing the precision when the system assigns a landing area. In Table 1, Table 2 and Table 3, the residual planes technique demonstrates high performance in recall but low performance in other metrics. This is because the component analysis is limited to evaluating whether a surface is flat. In addition, unlike the proposed approach, PCA does not consider landing zone accessibility.

The Support Vector Machine (SVM) is a widely used technique in learning algorithms, particularly when there is insufficient data to use deep learning techniques. The SVM uses the “kernel trick” to map data into a higher dimensional feature space. Although this technique works well for landing zone detection, it shows poor performance in terms of precision metrics (see Table 1, Table 2 and Table 3). This is problematic because misclassifying a landing zone may cause the UAV to land in areas that are inaccessible to the operator.

Logistic Regression is a machine learning technique commonly used for binary segmentation tasks. This technique has demonstrated its efficiency as an alternative when the characteristics exhibit a sufficiently linear relationship. However, this technique may exhibit a poor performance with a nonlinear or high-dimensional datasets. In our experiments, it proved to have the ability to detect flat areas for landing tasks but presented a lower accuracy compared to our proposed approach (see Table 1, Table 2 and Table 3). This is because our methodology is highly efficient for training high-dimensional datasets.

YOLOv7 is a deep learning technique optimized for computational processing efficiency. This efficiency is crucial for real-time applications, such as detecting landing zones for UAVs. We trained YOLOv7 using RGB images, unlike our proposed methodology, which uses only depth images. Despite its high performance in different previous applications, this architecture demonstrated poor performance in our experiments owing to changes in lighting conditions, scenario diversity, shadows, and other elements that typically impact detection accuracy (see Table 1). Experiments with synthetic images (see Table 2) demonstrate that YOLOv7-RGB performs better in controlled lighting and well-defined colors and textures. In addition, experiments with the ITTG dataset (see Table 3) improved its performance because the low height of the UAV does not visualize a great diversity of scenarios (corridors, green areas, trees, and small buildings).

We evaluated the efficiency of the YOLOv7 network using depth information. In our experiments, YOLOv7-Depth proved to be the least efficient technique on average (see Table 1, Table 2 and Table 3). In this case, depth information has fewer features (color, texture, and edges) that the YOLOv7 network usually extracts in its convolutional layers to identify complex objects. In contrast, the proposed methodology simplifies the identification of possible landing zones by partially analyzing depth image information using superpixels.

The CNN approach performs worse than the proposed approach since the convolutional processes need information with the most features. However, a depth image lacks essential features such as color and texture, which makes it challenging to identify objects or specific areas. Our approach, DNN-Superpixel Net, utilizes a fully connected network due to this nature of the data. This method processes a small sample of the area under analysis along with eight adjacent regions without essential features (color and texture).

Our approach (DNN-Superpixel Net) demonstrated the highest efficiency in terms of precision, IOU, and F-score metrics (see Table 1, Table 2 and Table 3). The precision achieved by the proposed method was 11.04% higher than that of the compared approaches. Thus, the proposal demonstrates that it more precisely identifies landing zones in the ground truth. This high performance is possible because the proposed methodology can generate large training sets from a few images. In this case, our approach can convert a few images to a big dataset. For example, we can obtain approximately 43,750 training landing zones on 50 images; that is, we can obtain 875 training landing zones using an image (1024 × 1024 pixels). This dataset allows us to train a DNN and use the abstraction power of deep learning to achieve robust performance for complex tasks such as segmentation landing zones. This approach also takes into account the accessibility of the landing zones, making it useful for tasks like deliveries, rescue operations, and emergency landings. Figure A7, Figure A8, Figure A9 and Figure A10 show some qualitative results of the six semantic segmentation approaches for landing zones.

Figure 7, Figure 8 and Figure 9 show graphic PAM evaluations of different semantic segmentation approaches using the VPAIR [37], VALID [38], and ITTG datasets. This metric represents an overall measure of performance, calculated from six key indicators: specificity (SP), sensitivity (SE), Jaccard index (JI), accuracy (AC), AUC, and F1-score. Each plot shows a polygon where the blue shaded area represents the PAM score. A larger shaded area indicates a better general performance. In these experiments, our DNN-Superpixel Net model exhibits the largest shaded area in the three different datasets reaching PAM scores of 0.8823 in the VPAIR dataset [37], 0.9229 in the VALID synthetic dataset [38], and 0.9048 in the ITTG dataset. Other models, such as PCA, SVM, and Logistic Regression, outperform the proposed method in particular metrics in the tests with the different datasets. However, in terms of general performance, they show lower performance than the DNN-Superpixel Net. Architectures such as YOLO-RGB and YOLO-DEPTH exhibit smaller areas in the three datasets, reflecting lower performance. These results underline the effectiveness of DNN-Superpixel Net in semantic segmentation, demonstrating balanced and superior performance compared to other architectures.

Figure A1, Figure A2 and Figure A3 show graphic ROC-AUC evaluations of different semantic segmentation approaches using the VPAIR [37], VALID [38], and ITTG datasets. The receiver operating characteristic (ROC) curve is a graphical representation used to evaluate the performance of a binary classifier as its discrimination threshold is varied. It plots the true positive rate (TPR, sensitivity) against the false positive rate (FPR, 1-specificity) at different threshold values. The AUC (Area Under the Curve) is a measure derived from the ROC curve since it calculates the area under this ROC curve and thus quantifies the overall ability of the test to discriminate between positive and negative classes. The models SVM and Logistic Regression show a high performance with a near-top ROC-AUC score. However, the DNN-Superpixel Net model exhibits the highest performance in the three different datasets.

Figure A4, Figure A5 and Figure A6 show confusion matrices of different semantic segmentation approaches using the VPAIR [37], VALID [38], and ITTG datasets. The matrices show results for Principal Component Analysis (PCA), Support Vector Machine (SVM), Logistic Regression, YOLO-RGB, YOLO-DEPTH, and our proposal (DNN-Superpixel Net). Each matrix provides information about the classification performance by showing the sums of true positives, true negatives, false positives, and false negatives. These results are essential for selecting the most suitable model based on specific performance criteria shown in Table 1, Table 2 and Table 3.

#### Processing Time

The experimental results are encouraging because our approach (DNN-Superpixel Net) performs well in both real-world and synthetic scenarios. The fastest approaches were YOLOv7-RGB and YOLOv7-Depth. However, in our experiments, these approaches present an average of 16.92% and 18.44% lower performance, respectively, than our proposed methodology (see Table 4). PCA is the slowest approach because it uses region-growing algorithms to evaluate the zones. In contrast, our approach uses superpixel segmentation on depth images to identify and delineate zones faster. In addition, our approach is 3.7156 times faster than PCA. For example, this approach can process three images and PCA, with one image per cycle. The Logistic Regression and SVM approaches have processing times similar to our approach; however, their performance is lower than that of our approach. In addition, we compared the processing time of landing zone identification using depth imaging. In our experiments, we used an Intel Core i9 CPU, NVIDIA GeForce RTX 4070 GPU, ADATA 64GB RAM, and Intel Z790G ATX Gaming motherboard.

Our method is faster and more accurate than PCA, as it simplifies the detection of zones in the depth image through superpixel segmentation. Subsequently, we analyze the zone boundaries and, through a DNN, determine whether it is accessible to a person and suitable for landing. In contrast, the PCA method identifies flat areas using a region-growing algorithm applied to a 3D point cloud, which entails a higher computational cost. In addition, PCA does not evaluate the flat zone environment, which may lead to selecting unsuitable locations, such as building rooftops, for landing. While PCA can be advantageous in emergency landings to minimize damage to the UAV, our approach is more effective for delivery tasks, as it provides a faster and more accurate response.

While YOLOv7 is faster in landing zone detection, its efficiency is limited due to its approach based solely on visual features, which can lead to inaccurate selections in complex environments. In contrast, our DNN-Superpixel, although slower, offers higher accuracy by considering both superpixel segmentation and environment assessment through adjacency analysis. This capability allows the identification of areas accessible to a person, reducing the risk of selecting unsuitable areas for landing. Therefore, while YOLOv7 is advantageous in scenarios where speed is a priority, our DNN provides a more robust and reliable solution for applications requiring higher accuracy.

### 4.4. Three-Dimensional Landing Zone Segmentation Result

We compare and evaluate the centroids of (X, Y, Z) coordinates obtained by semantic segmentation of landing zones using six different approaches: PCA, SVM, Logistic Regression, YOLOv7-RGB, YOLOv7-Depth, and DNN-Superpixel Net. For that, we performed experiments with three datasets containing outdoor scenes with varying characteristics: the VPAIR [37], VALID [38], and ITTG datasets. The scenes in these datasets have maximum depths of 300 m for VPAIR, 50 m for VALID, and 27 m for the ITTG dataset, which introduces different levels of complexity into the experiments. The metric used to evaluate the performance of each approach was the RMS (Root Mean Square) error of the coordinates (X, Y, Z), measured in meters. For the VPAIR dataset, with a maximum depth of 300 m, an RMS (Z) of 1.5241 represents an absolute error of approximately 1 m and 52 cm. Thus, this highlights the direct relationship between error magnitude and scene depth. Figure A10 shows some qualitative results of our semantic segmentation from the analysis of the 3D structure of the scene. Table 5 shows the RMS error of the centroids estimated by each segmentation approach compared to the ground truth.

A correctly identified center of mass provides an ideal reference point for executing controlled or automatic descent maneuvers, allowing the vehicle to land in a balanced manner and reducing the risk of tipping or rolling over. The correct location of the center of mass not only facilitates a safe landing but also optimizes the performance of the vehicle’s control system by more accurately managing its interaction with the terrain. On the other hand, inaccurate detection of the center of mass can cause several serious problems. An error in its estimation can deflect the expected landing point, causing the vehicle to touch the ground in an unstable position. The above increases the risk of mechanical damage or rollover, especially on uneven or sloping terrain. Therefore, ensuring accuracy in the center of mass detection is essential for system integrity and reliability in complex operational scenarios for landing tasks.

## 5. Conclusions

This study introduced a new methodology for landing zone segmentation from depth information using a DNN-Superpixel approach. This methodology consists of three phases. First, the proposal involves clustering depth information using superpixels to segment, locate, and delimit zones within the scene. Second, we proposed feature extraction from adjacent objects through a bounding box of the analyzed area. Finally, this methodology used a Deep Neural Network (DNN) to segment an area as landable or non-landable, considering its accessibility. We consider that a landing zone is accessible when it is practical for a user to retrieve the drone easily, that is, free of obstacles and easily accessible to a person on foot.

The quantitative experiments involved identifying suitable landing zones. We used three datasets that provide different aerial outdoor environments (VPAIR [37], VALID [38], and ITTG datasets). Our semantic segmentation of landing zones achieved an average recall rate of 0.953; that is, 95.3% of the landing zones were identified in the ground truth. The landing zone detection achieved an average precision of 0.949, meaning that this approach segments 94.9% of the landing zones correctly. In addition, the proposal has an F-score of up to 0.951, which shows that our proposed configuration has high performance in identifying accessible landing zones. The proposal proved the best precision, accuracy, PAM, and F-score metrics compared with other approaches. The proposed semantic segmentation of landing zones is innovative since it allows us to locate and delineate areas using only depth images.

The results of this study indicate that the proposed methodology is feasible and promising. Our methodology considers the landing zone accessibility, which facilitates its application to multiple tasks; that is, we aimed to create a tool that can identify easily accessible landing zones in open outdoor environments. Examples of concrete applications are emergency landing, rescue, and package delivery.

Our methodology was evaluated in diverse environments and conditions to assess its generalizability. Initially, we conducted tests using UAV-acquired aerial imagery at an altitude of approximately 300 m, encompassing urban areas, green spaces, and forests. Additionally, we evaluated the approach with synthetic images captured at 50 m, enabling a comparative analysis across scenarios such as highways, airports, and buildings. Also, we generated a dataset with aerial images taken at 27 m, incorporating elements like green areas, trees, and corridors. This multi-altitude and multi-environment evaluation provided a comprehensive assessment of our methodology’s performance under varying terrain and contextual challenges.

The proposed methodology allows the evaluation and segmentation of accessible landing zones, ensuring that a UAV can land safely and without risk of damage. A key aspect of our proposal is segmentation using superpixels, which, combined with data extraction of each zone and its surrounding areas, enables the generation of a comprehensive and representative dataset. This approach optimizes the use of depth images, maximizing the information available for training deep learning models and improving accuracy in identifying safe landing zones. Although the results obtained have demonstrated the effectiveness of our proposal in different environments (urban and green areas), we recognize the importance of evaluating its applicability in other conditions, such as urban scenarios with higher density of obstacles or in UAV platforms with different sensor configurations. In future work, we will consider extending the validation to more diverse environments and explore adjustments in the methodology to improve its adaptability to new operational conditions.

Finally, we can conclude that our approach segments 3D landing zones into easily accessible areas using only depth information. In this study, we investigate how to identify suitable landing zones in outdoor environments when the camera provides a top-down view for autonomous aerial vehicles. In addition, this proposal enables the generation of a large dataset from only a few images, allowing the use of deep learning techniques for tasks such as identifying areas for UAV landing. This strategy combined the abstraction power of deep learning with the information provided by the clustering of superpixels; in our opinion, it brings the best of the two worlds to address the challenge of depth information interpretation. To our knowledge, the proposed approach is the first study to consider landing zone accessibility using deep learning with only depth information in aerial outdoor environments.

## Figures and Tables

**Figure 1 sensors-25-02517-f001:**
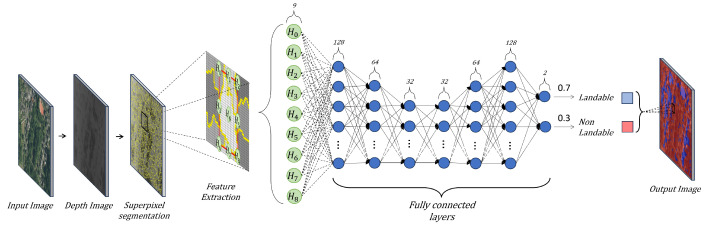
Block diagram of the proposed methodology.

**Figure 2 sensors-25-02517-f002:**
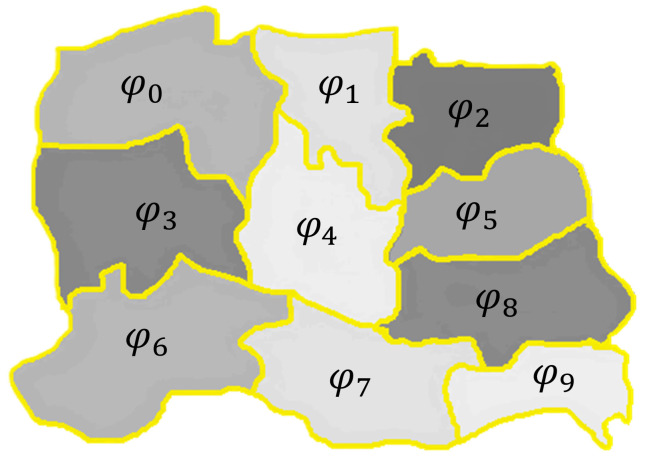
Region of the superpixel image. The gray sections are the superpixels $\phi_{i}$, and the yellow lines represent the boundaries of each superpixel.

**Figure 3 sensors-25-02517-f003:**
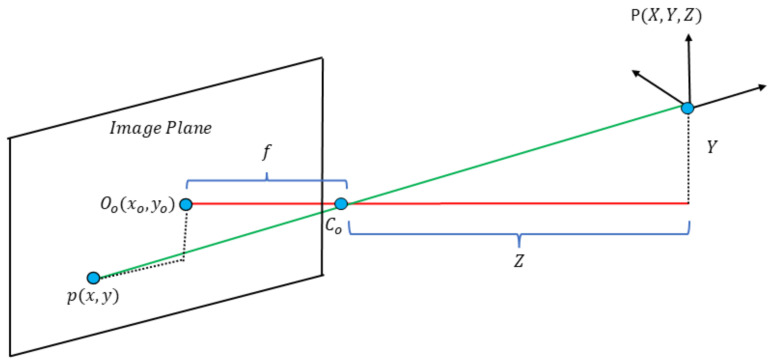
Pinhole camera geometry, where P(X,Y,Z) is a point in space, and p(x,y) is its projected point on the image plane. $C_{o}$ is the center of the camera or center of projection. $O_{o}$ is the origin of coordinates in the image plane. *f* is the distance between the center of the camera $C_{o}$ to the origin of coordinates of the image plane $O_{o}$.

**Figure 4 sensors-25-02517-f004:**
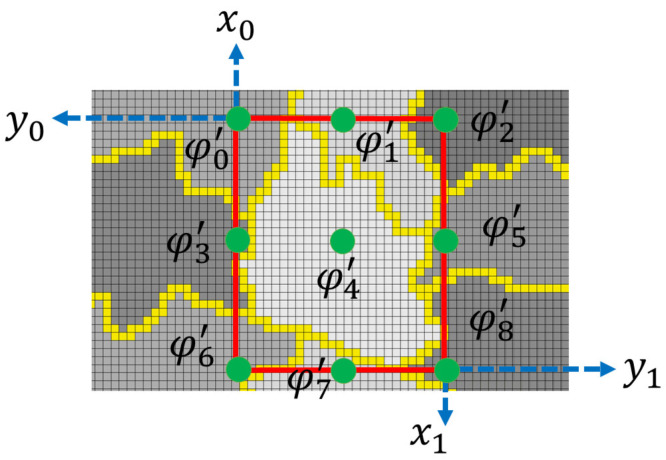
A region with nine superpixels with homogenized heights, a red bounding box, and green dots that represent the heights extracted in *H*.

**Figure 5 sensors-25-02517-f005:**
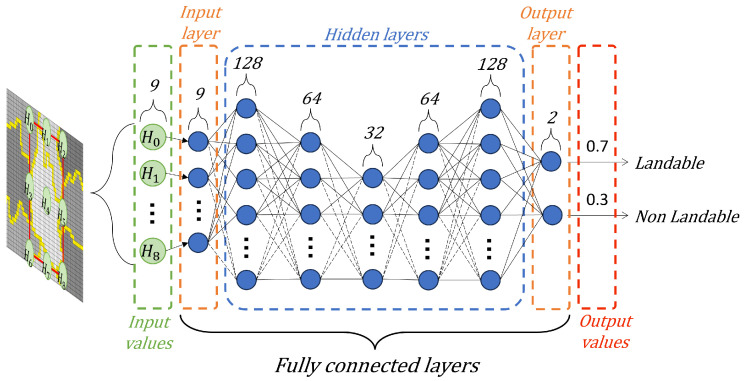
DNN proposal for zone classification.

**Figure 6 sensors-25-02517-f006:**
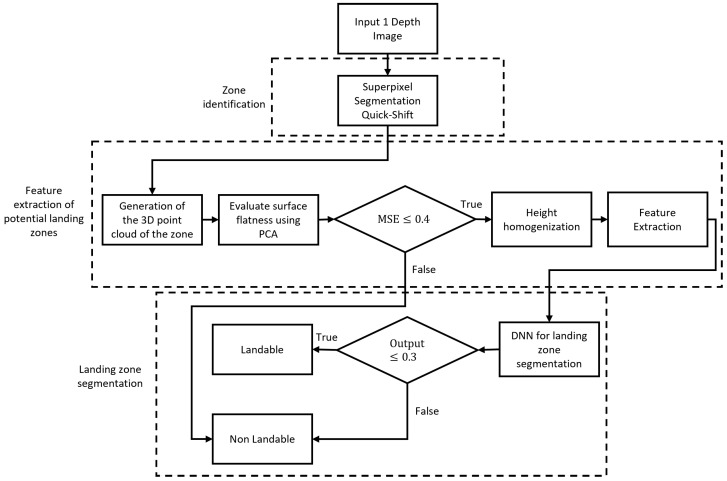
Diagram of the proposed methodology.

**Figure 7 sensors-25-02517-f007:**
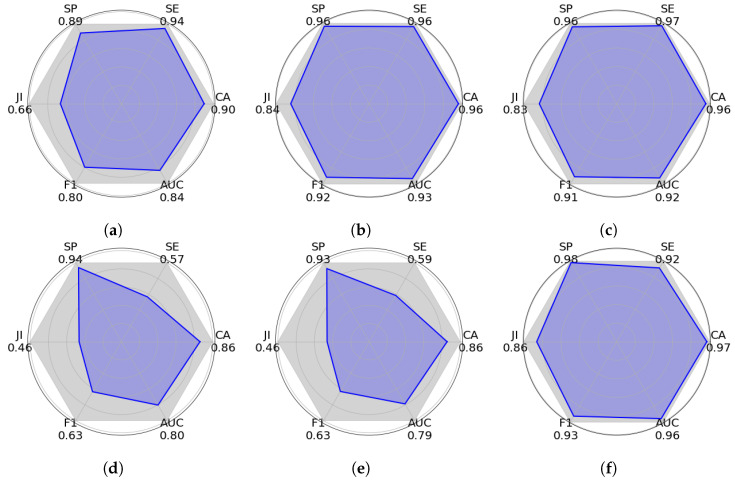
Comparative evaluation of machine learning models using Polygon Area Metric (PAM). The metrics were calculated using the experiments conducted on the VPAIR dataset [37]. The plots show six key performance indicators: specificity (SP), sensitivity (SE), Jaccard index (JI), accuracy (AC), Area Under the Curve (AUC), and F1-score, with PAM representing the blue shaded area inside each polygon. The larger polygon area indicates better overall performance according to PAM. (**a**) PCA (0.7208); (**b**) SVM (0.8728); (**c**) Logistic Regression (0.8667); (**d**) YOLO-RGB (0.5225); (**e**) YOLO-DEPTH (0.5182); (**f**) DNN-SP (0.8823).

**Figure 8 sensors-25-02517-f008:**
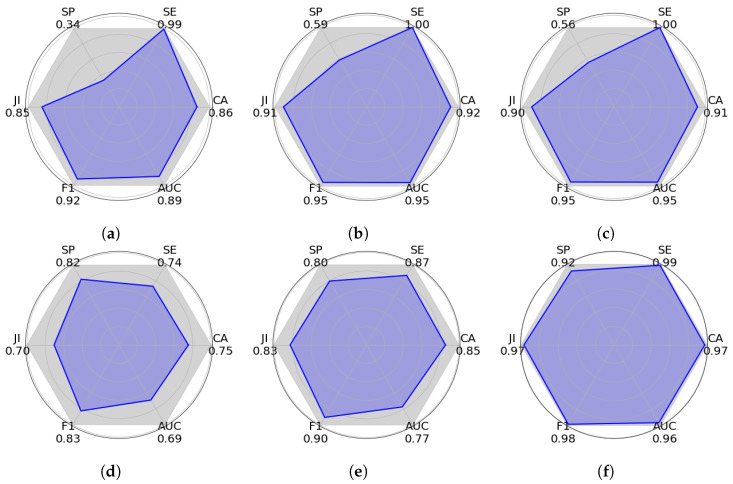
Comparative evaluation of machine learning models using Polygon Area Metric (PAM). The metrics were calculated using the experiments conducted on the VALID dataset [38]. The plots show six key performance indicators: specificity (SP), sensitivity (SE), Jaccard index (JI), accuracy (AC), Area Under the Curve (AUC), and F1-score, with PAM representing the blue shaded area inside each polygon. The larger polygon area indicates better overall performance according to PAM. (**a**) PCA (0.5787); (**b**) SVM (0.7484); (**c**) Logistic Regression (0.7304); (**d**) YOLO-RGB (0.5394); (**e**) YOLO-DEPTH (0.6710); (**f**) DNN-SP (0.9229).

**Figure 9 sensors-25-02517-f009:**
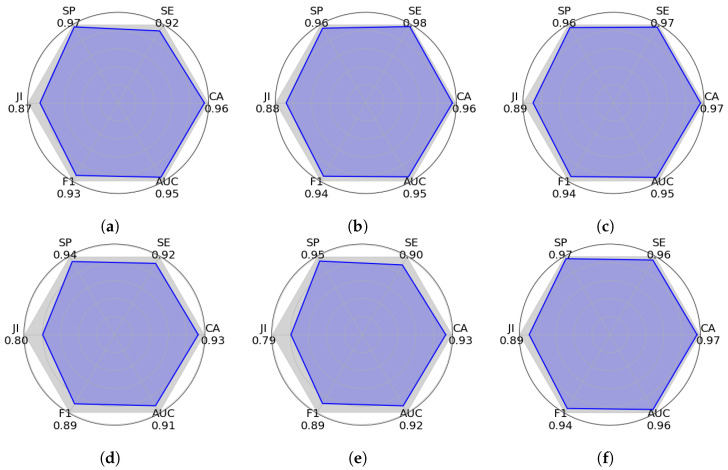
Comparative evaluation of machine learning models using Polygon Area Metric (PAM). The metrics were calculated using the experiments conducted on the ITTG dataset. The plots show six key performance indicators: specificity (SP), sensitivity (SE), Jaccard index (JI), accuracy (AC), Area Under the Curve (AUC), and F1-score, with PAM representing the blue shaded area inside each polygon. The larger polygon area indicates better overall performance according to PAM. (**a**) PCA (0.8788); (**b**) SVM (0.8973); (**c**) Logistic Regression (0.904); (**d**) YOLO-RGB (0.8172); (**e**) YOLO-DEPTH (0.8101); (**f**) DNN-SP (0.9048).

**Table 1 sensors-25-02517-t001:** Landable and non-landable zone segmentation on the VPAIR dataset [37]. We compare different semantic segmentation approaches using precision (PR), recall/sensitivity (RE/SE), F1-score (F1), accuracy (AC), specificity (SP), Jaccard index (JI), Area Under the Curve (AUC), and Positive Agreement Measure (PAM).

VPAIR Dataset [37]								
	**PR**	**RE/SE**	**F1**	**AC**	**SP**	**JI**	**AUC**	**PAM**
**PCA**	0.6898	0.9446	0.7973	0.8992	0.8871	0.6629	0.8367	0.7208
**SVM**	0.8748	0.9595	0.9152	0.9627	0.9635	0.8436	0.9319	0.8728
**L. Regression**	0.8566	0.9698	0.9097	0.9596	0.9569	0.8344	0.9242	0.8667
**YOLO-RGB**	0.7152	0.5691	0.6338	0.8620	0.9398	0.4639	0.8033	0.5225
**YOLO-Depth**	0.6824	0.5862	0.6306	0.8559	0.9275	0.4605	0.7882	0.5182
**CNN**	0.8519	**0.9850**	0.9099	0.9612	0.9552	0.8421	0.9247	0.8747
**DNN-S. Net**	**0.9339**	0.9170	**0.9254**	**0.9690**	**0.9828**	**0.8612**	**0.9560**	**0.8823**

Bold emphasis indicates the best value obtained in the experimental run.

**Table 2 sensors-25-02517-t002:** Landable and non-landable zone segmentation on the VALID synthetic dataset [38]. We compare different semantic segmentation approaches using precision (PR), recall/sensitivity (RE/SE), F1-score (F1), accuracy (AC), specificity (SP), Jaccard index (JI), Area Under the Curve (AUC), and Positive Agreement Measure (PAM).

VALID Synthetic Dataset [38]								
	**PR**	**RE/SE**	**F1**	**AC**	**SP**	**JI**	**AUC**	**PAM**
**PCA**	0.8566	0.9922	0.9194	0.8611	0.3395	0.8509	0.8864	0.5787
**SVM**	0.9066	**0.9998**	0.9509	0.9176	0.5902	0.9065	0.9528	0.7484
**L. Regression**	0.9004	**0.9998**	0.9475	0.9115	0.5603	0.9003	0.9497	0.7304
**YOLO-RGB**	0.9429	0.7359	0.8266	0.7533	0.8228	0.7045	0.6911	0.5394
**YOLO-Depth**	0.9446	0.8681	0.9047	0.8539	0.7974	0.8260	0.7738	0.6710
**CNN**	0.9730	0.9919	0.9822	0.9718	0.8889	0.9657	**0.9706**	0.9163
**DNN-S. Net**	**0.9791**	0.9873	**0.9832**	**0.9731**	**0.9163**	**0.9670**	0.9635	**0.9229**

Bold emphasis indicates the best value obtained in the experimental run.

**Table 3 sensors-25-02517-t003:** Landable and non-landable zone segmentation on our ITTG dataset. We compare different semantic segmentation approaches using precision (PR), recall/sensitivity (RE/SE), F1-score (F1), accuracy (AC), specificity (SP), Jaccard index (JI), Area Under the Curve (AUC), and Positive Agreement Measure (PAM).

ITTG Dataset								
	**PR**	**RE/SE**	**F1**	**AC**	**SP**	**JI**	**AUC**	**PAM**
**PCA**	**0.9361**	0.9226	0.9293	0.9592	**0.9742**	0.8679	0.9523	0.8788
**SVM**	0.9027	**0.9773**	0.9385	0.9628	0.9569	0.8841	0.9465	0.8973
**L. Regression**	0.9178	0.9700	0.9432	0.9661	0.9645	0.8925	0.9526	0.9040
**YOLO-RGB**	0.8633	0.9167	0.8892	0.9337	0.9406	0.8004	0.9141	0.8172
**YOLO-Depth**	0.8734	0.8970	0.8851	0.9324	0.9468	0.7938	0.9154	0.8101
**CNN**	0.8743	0.9762	0.9201	0.9482	0.9336	0.8465	0.9257	0.8642
**DNN-S. Net**	0.9340	0.9552	**0.9445**	**0.9674**	0.9724	**0.8948**	**0.9577**	**0.9048**

Bold emphasis indicates the best value obtained in the experimental run.

**Table 4 sensors-25-02517-t004:** Average processing times (seconds) for different semantic segmentation approaches. This comparison uses three datasets that provide different outdoor scenes (VPAIR [37], VALID [38], and our ITTG datasets).

Approach	VPAIR	VALID	ITTG
**PCA**	3.28	8.07	16.47
**SVM**	0.9392	2.2790	3.96
**Logistic Regression**	0.9259	2.2262	3.95
**YOLO-RGB**	0.0523	0.0595	0.0583
**YOLO-Depth**	**0.0471**	**0.0503**	0.0595
**DNN-Superpixel Net**	0.9454	2.2538	4.02

Numbers in bold indicate the best scores, the lower the better.

**Table 5 sensors-25-02517-t005:** Evaluation of 3D landing zone segmentation. We compare the centroids of the (X, Y, Z) coordinates of the landing zone segmentation with the ground truth. We analyze different approaches (PCA, SVM, Logistic Regression, YOLOv7-RGB, YOLOv7-Depth, DNN-Superpixel Net). In addition, we measure the RMS error in meters.

Dataset	Error	Approaches					
				**Logistic**	**YOLOv7-**	**YOLOv7-**	**DNN-Superpixel**
		**PCA**	**SVM**	**Regression**	**RGB**	**DEPTH**	**Net**
	RMS(X)	4.9031	2.1394	3.6114	5.0379	4.6755	**2.1**
VPAIR	RMS(Y)	8.4476	4.5299	5.7991	4.6323	5.8089	**2.6234**
	RMS(Z)	4.5261	1.1875	3.2211	**1.125**	1.3859	1.3923
	Ave. RMS	5.9589	2.6189	4.2105	3.5984	3.9568	**2.0386**
	RMS(X)	0.6632	0.9662	1.0599	7.0259	5.2428	**0.4032**
VALID	RMS(Y)	0.5467	0.7323	0.7035	2.6387	7.8879	**0.1366**
	RMS(Z)	0.7784	0.3422	0.3294	0.8817	0.6714	**0.0739**
	Ave. RMS	0.6628	0.6802	0.6976	3.5154	4.6007	**0.2046**
	RMS(X)	0.523	0.1604	0.1437	0.3571	0.5766	**0.1311**
ITTG	RMS(Y)	0.372	0.168	0.1441	**0.131**	0.3972	0.1433
	RMS(Z)	0.097	0.0594	0.0641	0.0851	0.0711	**0.0419**
	Ave. RMS	0.3307	0.1293	0.1173	0.1911	0.3483	**0.1054**

The numbers in bold indicate the best results in terms of RMS obtained for the three data sets (VPAIR, VALID, and ITTG). The smaller, the better.

## Data Availability

The datasets presented in this article are not readily available because the data are part of an ongoing study. Requests to access the datasets should be directed to corresponding autor.

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
