# Peer review of "Three-Dimensional Landing Zone Segmentation in Urbanized Aerial Images from Depth Information Using a Deep Neural Network–Superpixel Approach"

_sensors, 2025, doi:10.3390/s25082517_

Round 1
Reviewer 1 Report
Comments and Suggestions for Authors
1.The introduction provides a good overview of the problem and related work. However, it could benefit from a more detailed discussion of the limitations of existing methods and how the proposed approach specifically addresses these limitations.
2.The methodology is well-described, but the authors should consider adding more visual aids, e.g., flowcharts or diagrams, to help readers understand the different steps involved in the DNN-Superpixel approach.
3.Although the proposed method shows promising results, the computational complexity and processing time should be addressed more comprehensively. The manuscript mentions that the DNN-Superpixel Net is faster than PCA but slower than YOLOv7. However, a detailed analysis of the trade-offs between speed and accuracy would be beneficial for practical applications.
4.The study relies on specific datasets and environments. While the results are encouraging, it is unclear how well the methodology would generalize to other types of urban environments or different UAV platforms. The authors should provide more insights into the generalizability of their approach.
5.The conclusion should include a brief summary of the key findings and contributions of the study. Additionally, it would be helpful to outline potential future work, such as exploring other types of sensors or improving the generalizability of the methodology.

Reviewer 2 Report
Comments and Suggestions for Authors
This paper presents a novel technique for UAV landing zone segmentation using a DNN-Superpixel approach. The technique proposed consists of the subsequent data processing phases: (1) clustering depth information using superpixels to segment and delimit zones within the scene in view, (2) feature extraction from adjacent objects through a bounding box of the analyzed area, (3) using a Deep Neural Network to segment an area as landable or non-landable, considering its accessibility. The purpose is to make the landing zone free of obstacles and accessible for a user.
- The introduction includes relevant references for the article. Different approaches to selection of landing zone are observed and compared.
- The methods proposed are adequately described. The approach proposed is based on three phases of data processing: zone identification, feature extraction, segmentation of landing zone. These phases are described in details. The corresponding sections of the article are clear to understand. The methodology proposed combines traditional methods of image processing, such as Principal Component Analysis, with Deep Neural Network (DNN), developed and trained specially for automatic segmentation and classification of the possible landing zone. The DNN characteristics are considered in details.
- The experimental study of the proposed methods is performed carefully. The method was evaluated comprehensively using quantitative various evaluation metric recall, precision, specificity, jaccard index, accuracy, F1-score, Area Under the Curve. The results are clearly presented and illustrated by many figures and table data. The developed technique has been tested on different data sets. The tests show advantage of the method proposed in comparison with alternative methods, both traditional and based on artificial neural networks.
- All scientific information (research background, methods, experimental results, discussion, conclusion) is well written and clear to understand.
- The article is recommended for publication. All scientific sections should be published without revising.
- The minor revision concerns some formal sections, according the article template: “Funding”, “Institutional Review Board Statement” etc. These sections should be filled properly or excluded from the paper.
Reviewer 3 Report
Comments and Suggestions for Authors
This paper addresses the problem of autonomous landing zone detection for UAVs, proposing a deep neural network (DNN) and superpixel-based method for extracting 3D landing zones from depth information.
Novelty and Contributions
Strengths:
Innovative combination of DNN and superpixels: Unlike most existing landing zone detection methods, which rely on RGB vision or geometric point cloud analysis, this paper effectively integrates superpixel clustering with deep learning to improve detection accuracy.
Comprehensive experimental evaluation: The study utilizes multiple public datasets (VPAIR, VALID, ITTG) and compares the proposed approach with PCA, SVM, and YOLOv7, providing solid experimental support.
Areas for Improvement:
Comparison with state-of-the-art methods is insufficient: The paper only compares against PCA, SVM, and YOLOv7, but lacks comparisons with modern 3D semantic segmentation and Transformer-based approaches (e.g., PointNet++, Swin-Transformer). It is recommended to include a broader range of comparison methods.
Scientific Soundness of the Methodology
Strengths:
Well-structured methodology: The three-step framework (superpixel segmentation → feature extraction → DNN classification) is logically structured and well explained.
Feature extraction strategy is physically meaningful: Using Principal Component Analysis (PCA) for plane fitting and superpixel-based height analysis ensures the approach is suitable for UAV landing zone detection.
Superpixel clustering reduces computational complexity: Compared to per-pixel point cloud analysis, the superpixel-based approach effectively reduces the computational burden, which is a reasonable optimization strategy.
Areas for Improvement:
Unclear justification for the choice of superpixel algorithm: The paper selects Quick-Shift segmentation, but does not provide a comparison with other superpixel algorithms such as SLIC or Felzenszwalb. It is recommended to evaluate different superpixel methods and justify the selection.
DNN architecture is overly simplistic: The fully connected network. The paper does not explain why this architecture was chosen nor does it compare it with deeper networks or alternative architectures (CNN, Transformer, etc.).

Reviewer 4 Report
Comments and Suggestions for Authors
The presented study is certainly interesting. The authors have done a lot of work and presented interesting practically significant results.
However, the concept of precision (lines 545 - 547) needs to be clarified. This is due to the fact that when we talk about precision, it is usually characterised by the error, which is expressed in units of the measured value or percent. According to international standards, precision can also be characterised by uncertainty, which is estimated by probability.
In addition, attention should be paid to the design:
(a) Proofread the paper and remove typos, e.g. on lines 568-600;
(b) Add a reference to the annex in the text of the paper.
